# Feasibility of Algorithm-Based Clinical Decision Support for Suspected Urinary Tract Infections in Nursing Home Residents

**DOI:** 10.3390/antibiotics11101276

**Published:** 2022-09-20

**Authors:** Garrett P. New, Arif Nazir, Penny Logan, Christine E. Kistler

**Affiliations:** 1Gillings School of Global Public Health, University of North Carolina, Chapel Hill, NC 27599, USA; 2SHC Medical Partners, 12201 Bluegrass Pkwy, Louisville, KY 40299, USA; 3Department of Family Medicine, School of Medicine, University of North Carolina, Chapel Hill, NC 27599, USA; 4The Cecil G. Sheps Center for Health Services Research, University of North Carolina, Chapel Hill, NC 27599, USA

**Keywords:** antimicrobial stewardship, clinical decision support, urinary tract infections, nursing homes, long-term care

## Abstract

Urinary tract infections (UTIs) are commonly suspected in nursing home (NH) residents, commonly resulting in antimicrobial prescriptions, even when symptoms are non-specific. To improve the diagnosis and management of suspected UTIs in NH residents, we conducted a pilot test of a paper-based clinical algorithm across NHs in the southern U.S. with ten advanced practice providers (APPs). The paper-based algorithm was modified based on the clinical care needs of our APPs and included antimicrobial treatment recommendations. The APPs found the UTI antimicrobial stewardship and clinical decision support acceptable. The educational sessions and algorithm improved baseline confidence toward UTI diagnosing and treatment. The APPs thought the algorithm was useful and did not negatively impact workload. Feedback from the pilot study will be used to improve the next iteration of the algorithm as we assess its impact on prescribing outcomes.

## 1. Introduction

Urinary tract infections (UTIs) are the most diagnosed nursing home (NHs) infections, accounting for significant overprescribing in this setting [1,2]. The Centers for Disease Control and Prevention state that diagnostic testing and management algorithms may help clinicians differentiate asymptomatic bacteriuria from symptomatic UTIs, reducing inappropriate antimicrobial use [3]. Unfortunately, most antimicrobial algorithms fail to consider end-user workflow, limiting these interventions’ impact [4]. To improve NH antimicrobial stewardship, a UTI diagnostic algorithm was developed from international Delphi consensus guidelines [5]. This algorithm is undergoing validation in the Netherlands but, to our knowledge, has not been tested in the U.S. [6]. Our goal is to understand advance practice providers’ (APPs) attitudes toward clinical decision support and seek end-user feedback to improve the feasibility of a clinical decision support tool prior to full-scale testing. 

## 2. Methods

Based on feedback from our NH colleagues, we modified the published algorithm to highlight the multiple entry points into the clinical decision-making pathway and specified empiric antimicrobial treatment options (Figure 1) based on current Infectious Diseases Society of America guidelines [7]. The algorithm was initially developed using a modified-Delphi process of international experts for the Dutch Association of Elderly Care Physicians to identify the signs and symptoms associated with a UTI in an NH resident [5]. These signs and symptoms were then incorporated into a flow chart for deciding when an antibiotic is indicated for a suspected UTI in NH residents based on the final Delphi consensus. It also provides a list of the non-specific signs and symptoms which are often falsely attributed to the presence of a UTI but should prompt evaluation for other causes. We iteratively met for several weeks, modifying the flow in the algorithm to best align with clinical care based on feedback from our nursing home colleagues. 

We conducted a feasibility study on a group of 20 APPs providing NH primary care in a large NH chain across the southern U.S. Our goal was to assess the effect of a UTI educational session and the performance of the modified UTI algorithm. The first half of the training session consisted of a 30-min educational session on the current evidence for the diagnosis and management of UTIs in NH residents. We had 18 APPs complete both a pre-training and a post-training survey assessing attitudes toward UTIs and clinical decision support, confidence toward UTI diagnosis and treatment, and APP demographics. 

Half also were randomly selected to receive an additional 30-min training session on the paper-based version of the algorithm that explained the rationale and clinical decision support tool. Our original study called for us to examine actual antibiotic prescribing patterns between each half of the sample, but due to COVID, these data could not be collected. We collected their reported prescribing decisions based on the algorithm in the 10 APPs randomized to the intervention. The intervention APPs provided written feedback regarding algorithm usefulness and potential changes to the algorithm for each suspected UTI case. They also provided feedback regarding workload via the NASA Task Load Index (NASA-TLX) [8]. After one month, intervention APPs completed a post-intervention survey assessing the same pre-training attitudes. The Institutional Review Board (IRB) at the University of North Carolina at Chapel Hill conducted an exemption review of our submission to determine whether the proposed study needed a full ethical review. However, given that the research was a benign behavioral intervention of health care professionals using survey data, the study was determined to be exempt from needing full ethical review. 

Quantitative survey data were analyzed using STATA and Microsoft Excel software. One-sample 95% confidence intervals were calculated for each question in the pre-training, post-training, and post-intervention surveys. Due to aggregated and de-identified data, we were unable to conduct within-group comparisons (i.e., paired *t*-test) to determine the significance of the surveys, and all between-group comparisons were nonsignificant.

## 3. Results

Of the 18 APPs who responded, the mean age was 44 years (SD 8.8 years), with an average of 3–5 years in practice; 61% worked in an urban setting, and 100% were female. 

Advance practice providers’ attitudes toward UTI antimicrobial stewardship and clinical decision support were favorable for all questions in the pre-training survey. These attitudes remained favorable in both the post-training for all participants and the post-intervention survey for the ten advance practice providers after one month of using the algorithm. When asked in the post-intervention survey about willingness to try guideline-assisted prescribing even if it required a decision support aid (interval scale ranging from “0-not at all” to “4-a very great extent”), the mean response was 3.1 (95% CI 2.8–3.4) (Table 1). When asked if participation in the training session addressed knowledge and skills that can be used in practice (interval scale ranging from “0-strongly disagree” to “4-strongly agree”), the mean response was 3.0 (95% CI 2.2–3.8). The mean response was 3.1 (95% CI 2.3–3.8) concerning the effectiveness of the design of the training session. Post-intervention, the mean response (interval scale ranging from “0-strongly disagree” to “4-strongly agree”) for whether there was adequate time to complete the algorithm was 3.0 (95% CI 2.2–3.8). 

The APPs were confident in their ability to diagnose and treat UTIs in both the pre- and post-training surveys and the post-intervention survey (all scores were below 2 (interval scale ranging from “1-very confident” to “4-not very confident”). Because we had aggregated and de-identified data, we were unable to conduct within-group comparisons (i.e., paired *t*-test) to determine the significance of the surveys, and all between-group comparisons were nonsignificant. When asked their confidence in determining a UTI in a nursing home resident in the post-training survey, the mean response was 1.6 (95% CI 1–2.2), and their confidence in exploring diagnoses other than UTIs for non-specific signs and symptoms had a mean response of 1.8 (95% CI 1.4–2.1). When asked if it was appropriate to reduce unnecessary antimicrobial use for UTIs (interval scale ranging from “1-inappropriate” to “7-appropriate”), the mean response was 6.6 (95% CI 6.2–7.1).

The algorithm was used in 23 suspected UTI cases over a month. The APPs reported the algorithm helped in 91% of cases. Four antibiotic prescribing decisions reported by the APPs did not appear to follow the algorithm. Based on their report on patients’ symptoms, three would have fallen under empirical therapy and one under active monitoring. The two most common antibiotics prescribed were TMP/SMX and nitrofurantoin. Collected in 19 of 23 uses, the mean NASA-TLX workload (interval scale ranging from “1-very low” to “20-very high”) was 2.4. Seven changes to the algorithm were suggested, including additional diagnostic guidance for residents unable to report symptoms due to dementia and expanded antimicrobial options.

## 4. Discussion

To our knowledge, this was one of the first uses of a suspected UTI clinical decision support algorithm in real-time clinical care. Other UTI clinical decision support is being developed but has not been examined in clinical trials [9,10]. The recently published results of the original UTI algorithm in the Netherlands demonstrated improvements in appropriate antibiotic prescribing but were not statistically significant [6]. These findings were underpowered but speak to the need for prescribing nudges or other decision support features not currently in UTI algorithms [9]. Reassuringly, our NH APPs’ attitudes towards antibiotic stewardship and the use of guideline algorithms were positive. Though underpowered and lacking generalizability, our results suggest that the algorithm was feasible among APPs in our NHs.

While several UTI clinical decision tools have shown benefits, it is unclear what components contribute the most to their success [11]. As opposed to the work by Petterson et al., our educational sessions were supported by an actual clinical decision support algorithm [12]. As opposed to work by Nace et al. and Pasay et al., we targeted only prescribers and not nursing staff or families [13,14]. For our intervention, we modified the published algorithm to highlight the multiple entry points into the clinical decision-making pathway. We also specified empiric antimicrobial treatment options. We added the first-line empiric treatment options based on current Infectious Diseases Society of America guidelines for UTI in NH residents [7,15]. Our modified version is paper-based instead of in the the electronic health record as is the Dutch algorithm, which is part of our development process. The final version instructed clinicians to start at one of three symptom categories and then follow the steps of the algorithm to determine if the resident needs empiric therapy, targeted therapy based on cultures, or active surveillance. Because the algorithm was intended to support clinical decision-making and not supplant it, we included warnings that it should be used in addition to clinical judgment.

Our study was limited by several factors. Primarily, we were unable to look at actual antibiotic prescribing due to COVID-19 closing most nursing homes in the United States for research in 2020. We also only piloted the clinical decision support in one nursing home chain and in 10 advance practice providers, significantly limiting the generalizability. Once the clinical decision support tool is finalized, we will conduct a larger study powered to examine the efficacy of the tool. 

Future work will include adding decision support nudges into the intervention, antimicrobial tailoring for the empiric treatment options based on susceptibility rates of common UTI organisms per the NH’s antibiogram, if available, and noting other considerations when selecting antimicrobials such as dementia status, resident colonization, previous infections, and prior antimicrobial exposure in the previous three months.

## 5. Conclusions

Results from a pilot study of a clinical algorithm for the diagnosis and treatment of suspected UTIs in nursing home residents show that it is feasible in clinical care. These findings are relevant to nursing home providers interested in building their antimicrobial stewardship efforts.

## Figures and Tables

**Figure 1 antibiotics-11-01276-f001:**
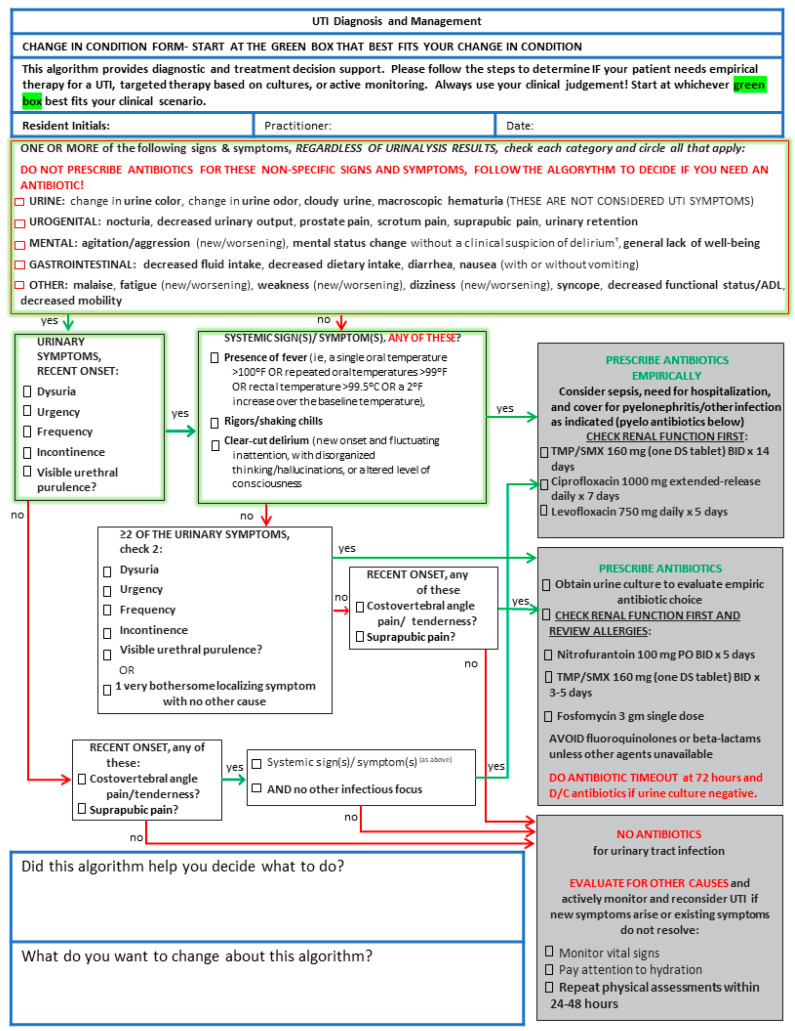
Clinical decision support algorithm for UTI diagnosis and management.

**Table 1 antibiotics-11-01276-t001:** Attitudes Toward UTIs and Clinical Decision Support and Clinical Confidence.

**Attitude, Mean (95% CI) ***	**Pre-Training Survey, *n* = 18**	**Post-Intervention Survey, *n* = 8**
I like to use new therapy/interventions to help my patients.	2.9 (2.5, 3.3)	3.1 (2.8, 3.4)
I am willing to try guideline-assisted prescribing even if I have to use a decision support aid.	3.1 (2.7, 3.5)	3.1 (2.8, 3.4)
I know better than academic researchers how to care for my patients.	1.4 (1.1, 1.8)	1.4 (0.6, 2.1)
I am willing to use new and different types of guideline-assisted prescribing decision support aids developed by researchers.	3.2 (2.9, 3.4)	3.0 (2.6, 3.4)
Guideline-assisted prescribing decision support aids are not clinically useful.	0.7 (0.2, 1.2)	1.0 (0.0, 2.0)
Clinical experience is more important than using guideline-assisted prescribing decision support aids.	1.5 (1.2, 1.8)	1.8 (1.2, 2.3)
I would not use guideline-assisted prescribing decision support aids.	0.7 (0.2, 1.2)	1.1 (−0.2, 2.4)
I would try a new decision support aid even if it were very different from what I am used to doing.	2.8 (2.5, 3.2)	2.8 (2.2, 3.3)
**Confidence, Mean (95% CI) ****	**Pre-Training Survey,** * **n** * **= 18**	**Post-Intervention Survey,** * **n** * **= 8**
Determining whether or not a nursing home rent has a urinary tract infection.	1.7 (1.3, 2)	1.6 (1, 2.2)
Exploring diagnoses other than UTIs for non-specific signs and symptoms.	2.1 (1.7, 2.4)	1.8 (1.4, 2.1)
Recommend non-pharmacologic practices to prevent the development of a UTI.	2.2 (1.7, 2.6)	1.9 (1.3, 2.4)
Prescribing the appropriate antibiotic regimen for pyelonephritis.	2.1 (1.7, 2.5)	1.6 (1.2, 2.1)
Prescribing the appropriate antibiotic regimen for a simple cystitis/UTI.	1.9 (1.6, 2.3)	1.8 (1.2, 2.3)

* Interval scale ranging from not at all (0) to a very great extent (4); ** Interval scale ranging from very confident (1) to not very confident (4).

## Data Availability

Data available on request.

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
