# Peer review of "Feasibility of Algorithm-Based Clinical Decision Support for Suspected Urinary Tract Infections in Nursing Home Residents"

_antibiotics, 2022, doi:10.3390/antibiotics11101276_

Round 1

Reviewer 1 Report

This paper is developed within the framework of urinary tract infections. To improve the diagnosis and management of urinary tract infections in nursing home residents, in this paper, the authors try to present a pilot test of a paper-based clinical algorithm across nursing homes in the southern U.S with eight advanced practice providers.

After a detailed review, I consider that this looks like as an early draft. The whole document must be revised in deep, introducing the topic in a coherent and cohesive manner. With the explanations given it is impossible to follow and evaluate your proposal.

For that reason, I propose to reject the paper with the option of resubmission in the future, once the paper is ready.

Please, see my comments in the attached pdf file.

Author Response

Main concerns

• I think that the authors should restructure de paper. Why does the section of results is after the Introduction? Why Materials & Methods is the last section? I consider that the right organization should be Introduction, Materials & Methods, Results and Discussion.

Response: We greatly appreciate this comment.  The journal itself asked for this particular format. I have never formatted a paper the way it was requested, as I traditionally do Intro, Methods, Results, Discussion, Conclusion, but this was specifically requested by this journal.  I have reformatted it as you proposed.  If it is accepted, I am happy to reformat per their specifications.

• According to the instructions for authors: ‘[…] all manuscripts must contain the required sections: Author Information, Abstract, Keywords, Introduction, Materials & Methods, Results, Conclusions, Figures and Tables with Captions, Funding Information, Author Contributions, Conflict of Interest and other Ethics Statements. Check the Journal Instructions for Authors for more details.’ Please, add also a conclusions section.

Response: We apologize for this oversight and have added in a final section with our conclusions. It now reads:

Results from a pilot study of a clinical algorithm for the diagnosis and treatment of suspected UTIs in nursing home residents show that it is feasible in clinical care. These findings are relevant to nursing home providers who are interested in building their antimicrobial stewardship efforts.

• I consider that the authors must introduce the algorithm and the clinical decision support system used. I consider that Figure 1 should be explained in detail.

Response: We appreciate the need for further details about the tool.  We have added details to the methods section that now reads:

The algorithm was initially developed using a modified-Delphi process of international experts for the Dutch Association of Elderly Care Physicians to identify the signs and symptoms associated with a UTI in a NH resident. These signs and symptoms were then incorporated into a flow chart for deciding when an antibiotic is indicated for a suspected UTI in NH residents based on the final Delphi consensus. It also provides a list of the non-specific signs and symptoms which are often falsely attributed to the presence of a UTI but should prompt evaluation for other causes.

• The Institutional Review Board is not shown in the text. It is said: ‘This research was conducted deemed exempt from review by the Institutional Review Board at the ##############.’. Later, it is said ‘Institutional Review Board Statement: Ethical review and approval were waived for this study due to its conduct in health care practitioners with de-identified data for quality improvement purposes.’.

Response: We realize that this is confusing.  We have modified this section for clarity. It now reads:

The Institutional Review Board (IRB) at the ############### conducted an exemption review of our submission to determine whether the proposed study needed a full ethical review. However, given that the research was a benign behavioral intervention of health care professionals using survey data, the study was determined to be exempt from needing full ethical review. 

• With the given explanations it is impossible to understand what was done. The problem should be clearly stated, and then presenting what are the objectives. The algorithm should also be clearly introduced and explained. After that, you should explain what you did in a comprehensive and coherent manner.

Response: We greatly appreciate this point we have reorganized the manuscript as suggested. We have added a line at the introduction to more clearly state our objectives. As noted above we have further explained the algorithm. We have added additional content to the methods. as well. The last sentence of the introduction reads:

This manuscript will present the results of an attitudinal survey of clinical decision support and a pilot study of a paper-based clinical decision support tool. 

• The results must be discussed in more detail in the discussion section, explaining and detailing its scope. The discussion section should clearly determine what is the main contribution of the article compared to other studies or similar works in the related field of study. Authors need to pay special attention to this comparison and highlight the relevance of their contributions

Response: We have updated the results section throughout to more explicitly explain where the results come from and we have added a section to the discussion to explain our changes and how it compares to work in the field which reads:

While several UTI clinical decision tools have shown benefit, it is unclear what components contribute the most to their success.11  As opposed to work by Petterson et al., our educational sessions were supported by an actual clinical decision support algorithm.12 As opposed to work by Nace et al and Pasay et al, we targeted only prescribers and not nursing staff or families.13,14 For our intervention, we modified the published algorithm to highlight the multiple entry points into the clinical decision-making pathway. We also specified empiric antimicrobial treatment options. We added the first-line empiric treatment options based on current Infectious Diseases Society of America guidelines for UTI in NH residents.7,15 Our modified version is paper-based as opposed to the Dutch algorithm as part of the development process. The final version instructed clinicians to start at one of three symptom categories and then follow the steps of the algorithm to determine if the resident needs empiric therapy, targeted therapy based on cultures, or active surveillance. Because the algorithm was intended to support clinical decision-making and not supplant it, we included warnings that it should be used in addition to clinical judgement..        

Other comments

• I think that the excessive use of the acronyms makes it difficult to follow the paper. See the case of the abstract.

Response: We appreciate this point and have removed APP from the manuscript and refer to them as advanced practice providers. We have removed NH from the manuscript and refer to them as nursing homes. We have left UTI as this is the most preferred term and universally used to describe urinary tract infections.

• I suggest you to add a paragraph at the end of the introduction explaining how the paper is organized.

Response We will be happy to add more about the organization of the manuscript to the introduction if directed by the journal.  We have reformatted to the current format as suggested but defer to the journal for final formatting.  We are happy to add a few sentences about this if needed.

• Revise the punctuation. It seems there are some problems.

Response: We are happy to revise the punctuation if there are problems. We have reread the manuscript and made several revisions throughout the text.

Reviewer 2 Report

This study outlines the importance of using algorithm in real-time clinical care and also, the importance of prescribers training seeing the therapeutic approach of a very frequent pathology, high consumer of antibiotics.

Although the material is very well structured, it should be noted that the conclusions section is missing.

Author Response

This study outlines the importance of using algorithm in real-time clinical care and also, the importance of prescribers training seeing the therapeutic approach of a very frequent pathology, high consumer of antibiotics.

Although the material is very well structured, it should be noted that the conclusions section is missing.

Response: We appreciate this. We have reformatted to help Reviewer 1 but will reformat after acceptance. We have added a conclusion section as noted above in response to Reviewer 1. Thank you for noting this.

Reviewer 3 Report

English language needs editing. Furthermore, Table needs to be numbered.

Intorduction is insufficient, so are the methods. Please expand both to include description of other alghoritms and to describe how the surveys were designed and what educational interventions consisted of and how was sample size determined.

You mention pre-educational survey, post-educational and post-intervention, only two groups of results are presented. Please revise

Discussion is missing limitations. Overall this study seems pointless in the way you present it. Wouldn't it have been better to have a control group and to look at the outcomes rather than opinions on use of alghoritm?

Author Response

• Intorduction is insufficient, so are the methods. Please expand both to include description of other alghoritms and to describe how the surveys were designed and what educational interventions consisted of and how was sample size determined.

Response: We have added additional descriptions in the introduction on the purpose of the manuscript. We have added additional information on the algorithm to the methods:

Based on feedback from our NH colleagues, we modified the published algorithm to highlight the multiple entry points into the clinical decision-making pathway and specified empiric antimicrobial treatment options (Figure) based on current Infectious Diseases Society of America guidelines.7 The algorithm was initially developed using a modified-Delphi process of international experts for the Dutch Association of Elderly Care Physicians to identify the signs and symptoms associated with a UTI in a NH resident.5 These signs and symptoms were then incorporated into a flow chart for deciding when an antibiotic is indicated for a suspected UTI in NH residents based on the final Delphi consensus. It also provides a list of the non-specific signs and symptoms which are often falsely attributed to the presence of a UTI but should prompt evaluation for other causes. We iteratively met for several weeks modifying the flow in the algorithm to best align with clinical care based on feedback from our nursing home colleagues.

We have this sentence to describe the education intervention:

The first half of the training session consisted of a 30-minute educational session on the current evidence for the diagnosis and management of UTIs in NH residents.

And we say in the next paragraph:

Half also were randomly selected to receive an additional 30-minute training session on the paper-based version of the algorithm that explained the rationale and clinical decision support tool.

This was a pilot study and so sample size was not considered. We have added a limitations section to the discussion to convey this:

Our study was limited by several factors. Primarily, we were unable to look at actual antibiotic prescribing due to COVID-19 closing most nursing homes in the United States for research in 2020. We also only piloted the clinical decision support in one nursing home chain and in 10 advance practice providers, significantly limiting the generalizability. Once the clinical decision support tool is finalized, we will conduct a larger study to powered to examine the efficacy of the tool.

• You mention pre-educational survey, post-educational and post-intervention, only two groups of results are presented. Please revise.

Response: As the post-intervention survey only included 10 individuals, we felt it more reasonable to include only the pre and post-training results. We have updated the methods and results to note which survey contained which results.

• Discussion is missing limitations. Overall this study seems pointless in the way you present it. Woudn't it have been better to have a control group and to look at the outcomes rather than opinions on use of alghoritm?

Response: We appreciate this point. We do agree that our manuscript is limited by the lack of actual prescribing data which was not available given the COVID-19 constraints for nursing home research. We have added a section to the discussion on the limitations of the study:

Our study was limited by several factors. Primarily, we were unable to look at actual antibiotic prescribing due to COVID-19 closing most nursing homes in the United States for research in 2020. We also only piloted the clinical decision support in one nursing home chain and in 10 advance practice providers, significantly limiting the generalizability. Once the clinical decision support tool is finalized, we will conduct a larger study powered to examine the efficacy of the tool.

Round 2

Reviewer 1 Report

All my doubts were answered. Thank you.